# Human Alpha 1 Antitrypsin Suppresses NF-κB Activity and Extends Lifespan in Adult *Drosophila*

**DOI:** 10.3390/biom12101347

**Published:** 2022-09-22

**Authors:** Ye Yuan, Max Van Belkum, Alana O’Brien, Alain Garcia, Karla Troncoso, Ahmed S. Elshikha, Lei Zhou, Sihong Song

**Affiliations:** 1Department of Pharmaceutics, College of Pharmacy, University of Florida, Gainesville, FL 32610, USA; 2L&L Biotechnology, Gainesville, FL 32610, USA; 3Department of Molecular Genetics & Microbiology, College of Medicine, University of Florida, Gainesville, FL 32610, USA; 4Genetics Institute, University of Florida, Gainesville, FL 32611, USA

**Keywords:** human alpha-1 antitrypsin (hAAT), senescence, NF-κB, SASP, *Drosophila*

## Abstract

Human alpha 1 antitrypsin (hAAT) is a multifunctional protein that has been shown to have anti-inflammatory and cellular protective properties. While previous studies demonstrated the antiaging potential of hAAT, the mechanism(s) underlying the antiaging effect remain elusive. In this study, we performed a detailed analysis of transcriptomic data that indicated that NF-κB-targeted genes and NF-κB-regulated pathways were selectively inhibited by hAAT treatment. We further showed that the first detectable impact of hAAT treatment was the inhibition of the nuclear activity of NF-κB. Subsequently, hAAT treatment suppressed the mRNA levels of NF-κB-targeted genes, as well as *NF-κB* itself (*P65* and *P50*), in human senescent cells. Using *Drosophila* models, we further examined the impact of hAAT on locomotor activity and endurance. Finally, using an adult-specific promotor, we demonstrated that overexpression of hAAT in the late stage of life significantly extended the lifespan of transgenic flies. These results extend the current understanding of the anti-inflammatory function of hAAT.

## 1. Introduction

Human alpha 1 antitrypsin (hAAT) has significant effects in multiple biological processes, including proteinase inhibition, tissue damage and repair, and apoptosis [1]. Recently, increasing efforts have been focused on the anti-inflammatory properties of hAAT beyond its protease inhibitor activity. Human AAT can suppress TNF-α-mediated upregulation of proinflammatory gene expression, including the expression of *TNF-α* itself, in human pulmonary vascular endothelial cells [2]. Human AAT can competitively bind to TNF-α receptors (TNFR1 and TNFR2), thereby blocking the interaction of TNF-α with its receptors [3]. Human AAT can also increase the production of anti-inflammatory factor IL-10 via activation of cAMP-mediated protein kinase A [4]. Although hAAT is a circulating protein, it can enter target cells, function intracellularly, and display an antiapoptotic effect [5]. In addition, hAAT can act as a scavenger of reactive oxygen species (ROS), which is a major inflammatory mediator and a marker of senescence [1,6]. As a potent inflammation regulator, hAAT has a potential protective function in a variety of inflammatory diseases, including rheumatoid arthritis [7,8,9], graft-versus-host disease [10], osteoporosis [11,12,13,14], inflammatory bowel disease [15], and type 1 diabetes [16]. With a reliable safety profile, hAAT has significant prospects for use in treating aging-related inflammatory diseases [17].

NF-κB is a family of transcription factors that plays a major role in stimulating inflammatory signaling upon activation. Increasing evidence suggests NF-κB-mediated inflammatory pathways contribute to aging and aging-related chronic inflammation. The suppression of NF-κB in the brain by transgenic inhibition of IKK-β significantly extended the lifespan of mice and improved aging-related cognition decline, muscle weakness, skin atrophy, bone loss, and collagen cross-linking [18]. The longevity proteins SIRT1 and SIRT6 can physically interact with p65/RelA and inhibit its transcription activity by deacetylating the p65/RelA protein or binding the promoter region of p65 [19,20,21]. In senescent cells, the senescence-associated secretory phenotype (SASP) can be inhibited by targeting RelA [22]. Also, therapy-induced senescence depends on NF-κB signaling [23]. It has been shown that hAAT treatment affected NF-κB nuclear translocation and DNA-binding activity [10,24]. Our group previously showed that transgenic expression of hAAT can suppress *Relish* (*Drosophila* ortholog of *NF-κB*)-mediated inflammatory pathways and extend the lifespan of *Drosophila* [25]. We also observed that hAAT can decrease the expression and secretion IL-6 in senescent human cells. However, the functional mechanism of hAAT in senescent cells remains to be further investigated.

In this study, we used an in vitro senescence model to study the effect of hAAT on NF-κB and NF-κB-induced SASP. As *Drosophila* has highly conserved regulatory pathways in innate immunity with vertebrates and advantages for developmental studies, we employed this model. In the *Drosophila* melanogaster model, we used a new expression system that is specific to the adult fat body (analogous to liver in mammals) to verify the anti-inflammaging effect of hAAT in adults by minimizing the complication of its potential impact on animal development.

## 2. Materials and Methods

### 2.1. Differentially Expressed Genes (DEGs), Enrichment Analysis, and Visualization

The RNA sequencing was performed as previously described [25]. The RNA sequencing data were analyzed using the DESeq2 method [26]. Multiple comparisons were corrected using the Benjamini–Hochberg (BH) procedure [27]. The gene expression fold-changes with an adjusted *p*-value less than 0.01 were identified as DEGs. The procedures mentioned above were performed using DESeq2 package in R 3.5.2. The Gene Ontology (GO) over-representation test was used for the enrichment analysis. Multiple comparisons were adjusted using the BH method. The enrichment analysis and related visualization were performed by utilizing the clusterProfiler package in R 3.5.2 [28]. Enrichr was adopted for the transcription factor and pathway enrichment analysis [29]. The Trust Transcription Factor (TTF) 2019 gene set, as well as Transcription Factor PPIs and the TRANSFAC and JASPAR PWMs gene sets, were used for the transcription factor enrichment analysis. The Kyoto Encyclopedia of Genes and Genomes (KEGG) 2019 gene set library was used for the pathway enrichment analysis. The *p*-value, adjusted *p*-value, and combined score were calculated following the method demonstrated by Chen et al. [30].

### 2.2. Cell Culture and X-Irradiation

The HCA2 cells were incubated in Eagle’s Minimum Essential Medium (MEM) (Sigma, St. Louis, MO, USA, Cat #: M4526) with 10% fetal bovine serum (FBS) (Atlanta Biologicals, Lawrenceville, GA, USA, Cat. No. S11150H), 1× penicillin/streptomycin, and 2 mM of L-glutamine.

The cells were subjected to 10 Gy (2.56 Gy/min) of X-irradiation using an XRAD 320 (Precision X-ray Inc., North Branford, CT, USA). Control cells were sham operated in parallel with the treated cells. The culture medium was immediately replaced with fresh medium following irradiation. The irradiated cells were treated with either 1× PBS or hAAT (Prolastin C^®^, Grifols, NC, USA; 2 mg/mL).

### 2.3. NF-κB Activity Assay

Nuclear extraction was performed with a Nuclear Extraction Kit (Cayman Chemical Company, Ann Arbor, MI, USA, Item No. 10009277) according to the protocol provided by the manufacturer. Then, the nuclear extraction was analyzed with an NF-κB (p65) Transcription Factor Assay Kit (Cayman Chemical Company, Ann Arbor, MI, USA, Item No. 10007889) according to the protocol provided by the manufacturer. The response reading was normalized to the protein concentration of the nuclear extraction.

### 2.4. Drosophila Lines

The LSP2-Gal4 and Yolk-Gal4 lines were obtained from the Bloomington stock center (Stock Nos. 6357 and 58814, respectively). The transgenic UAS-hAAT and UAS-GFP lines were generated by our group as reported previously [25]. These lines were backcrossed to the isogenic line w1118 (BDSC5905) 5–7 times to achieve a similar genetic background.

### 2.5. Lifespan Experiment

For the data presented in Figure 5D and Appendix A, flies were aged at a density of 12 flies per vial at 25 °C in a 12 h/12 h light/dark cycle at 40–60% humidity. Flies were transferred to fresh food medium with light CO_2_ anesthesia every 3–5 days and survival was recorded before being transferred to new food vials. Flies were cultured with Archon Scientific fly food #W10101 consisting of corn syrup/soy media.

### 2.6. Drosophila Activity and Endurance Tests

A locomotor activity test was performed using the TriKinetics *Drosophila* Activity Monitor (TriKinetics Inc, Waltham, MA, USA, Model: DAM5M) according to the protocol provided by manufacturer. A Python script was used to extract data for the statistical analysis. Endurance assays were performed as described previously [31]. Aged flies were transferred into empty vials and kept there for 15 min to acclimate them to the new environment. Flies were tapped down, following which they would reflectively climb up. This tap-down and climb procedure was repeated 4 times with a 1 min interval. For the fifth time, the number of flies that climbed to each height level were counted at the 2nd, 3rd, and 4th second in a recorded video.

### 2.7. ELISA Assay

For quantification of protein-level hAAT expression, 5 larvae and 6–8 adult flies were weighed and added into Zymo ZR BashingBead Lysis Tubes (Cat # S6003-50) containing 400 uL 1× proteinase inhibitor cocktail solution on ice. The flies were homogenized at 3500 rpm for 3 cycles and 30 s of rest between cycles using a BeadBug-6 homogenization machine. The beads were removed and the homogenate was spun down at 15,000 rpm for 5 min at 4 °C. The supernatant was then removed and again spun down at 15,000 rpm for 5 min at 4 °C. This second supernatant was harvested for further ELISA analysis.

### 2.8. RNA Extraction and Q-PCR

Fly and fibroblast RNA were extracted with a Quick-RNA MiniPrep (Genesee Scientific, San Diego, CA, USA, Cat #: 11-328) according to the protocol provided by the manufacturer. Fruit flies were homogenized at 3 cycles at 4000 rpm with 30 s of rest in between cycles with a BeadBug-6 homogenizer using 800 uL of DNA/RNA shield solution (Cat. No. R1100-250) in premade tubes containing 2.0 mm bashing beads provided by Zymo Research (Cat. No. S6003-50). Subsequent RNA extraction was done according to the Quick-RNA MiniPrep plus protocol from Zymo Research (Cat. No. R1051). The cDNA was prepared using reverse transcription of total RNA with a High-Capacity cDNA Archive Kit (Applied Biosystems, Foster City, CA, USA, Cat. No. 4368814). Q-PCR was performed with an ABI 7500 Fast thermocycler (Applied Biosystems, Foster City, CA, USA ) following protocols provided by the manufacturer. qPCRBio Sygreen Mix was used for qPCR according to the protocol provided by the manufacturer (Cat. No. PB20. 11-20). Technical triplicates were processed for each biological replicate.

### 2.9. Primers

The primer sequences used in this study were as follows: H-GAPDH forward: 5′-ATGGGGAAGGTGAAGGTCG-3′, H-GAPDH reverse: 5′-GGGGTCATTGATGGCAACAATA-3′, IL-6 forward: 5′-TACCCCCAGGAGAAGATTCC-3′, IL-6 reverse: 5′-TTTTCTGCCAGTGCCTCTTT-3′, p65 forward: 5′-GCATCCACAGTTTCCAGAAC-3′, p65 reverse: 5′-CACTGTCACCTGGAAGCAGA-3′, p50 forward: 5′-AACAGAGAGGATTTCGTTTCCG-3′, p50 reverse: 5′-TTTGACCTGAGGGTAAGACTTCT-3′, NF-κB2 forward: 5′-GTGCCTCCAGTGAGAA-3′, NF-κB2 reverse: 5′-AGGACACCCAGGTTGTTAAA-3′, CTSK forward: 5′-GTCTGAGAATGATGGCTGTGGA-3′, CTSK reverse: 5′-CATTTAGCTGCCTTGCCTGTTG-3′, cIAP1 forward: 5′-AGCCTGAGCAGCTTGCAAGTGC-3′, cIAP1 reverse: 5′-CCCATGGATCATCTCCAGATTCCC-3′, cIAP2 forward: 5′-CCGTCAAGTTCAAGCCAGTTACCC-3′, cIAP2 reverse: 5′-AAGCCCATTTCCACGGCAGC-3′, DIAP1 forward: 5′-GCGTGGAAATCGGTTGCTG-3′, DIAP1 reverse: 5′-GATGCGATCTAATGCTTCGGC-3′, DIAP2 forward: 5′-CACGCTTATGCAAGGTATGC-3′, DIAP2 reverse: 5′-GGGACAATTGGCTACACTGG-3′, D-GAPDH forward: 5′-AAGGGAATCCTGGGCTACAC-3′, D-GAPDH reverse: 5′-CGGTTGGAGTAACCGAACTC-3′, c-Rel forward: 5′-TTTTCCTGAGAGACCAAGACCT-3′, c-Rel reverse: 5′-GCTTGACTTGAAACCCCTGTAG-3′, IκB-α forward: 5′-GCTGAAGAAGGAGCGGCTACT-3′, IκB-α reverse: 5′-TCGTACTCCTCGTCTTTCATGGA-3′, IκB-β forward: 5′-GCTGACCTTGACAAACCGGA-3′, IκB-β reverse: 5′-GCCGGATTTCTCGTCCTCG-3′, IKK-β forward: 5′-AAATGAAAGAGCGCCTTGG-3′, IKK-β reverse: 5′-CACTGCTTGATGGCAATCTG-3′, Rpl32 forward: 5′-ATCGGTTACGGATCGAACAA-3′, Rpl32 reverse: 5′-GACAATCTCCTTGCGCTTCT-3′, DptA forward: 5′-ATTGGACTGAATGGAGGATATGG-3′, DptA: 5′-CGGAAATCTGTAGGTGTAGGT-3′ reverse, DptB forward: 5′-AGCCTGAACCACTGGCATA-3′, DptB reverse: 5′-AGATCGAATCCTTGCTTTGG-3′.

### 2.10. Statistical Analysis

The detailed number of samples and statistical methods used in the study are described in the Appendix A.

## 3. Results

### 3.1. Human AAT Treatment Specifically Inhibited NF-κB-Mediated SASP Gene Transcription and Inflammatory Pathways in Human Senescent Cells

A low dose of X-ray irradiation induces proliferation arrest and activates SASP in HCA2 cells (human foreskin fibroblasts), which is a widely accepted in vitro senescence model [32,33,34]. To test the effect of hAAT on the gene expression in senescent cells, HCA2 cells were cultured with culture medium plus hAAT or PBS for 5 days after the onset of senescence (Figure 1A). To exclude the potential interference of fetal bovine serum (FBS), senescent HCA2 cells were further cultured for 3 days with FBS-free medium (Figure 1A). At day 15, the mRNA was extracted and prepared for RNA-seq analysis. The RNA-seq data were originally analyzed using the Cufflink2 method and identified 42 DEGs (9 upregulated and 33 downregulated) [25]. The expressions of eight SASP genes were found to be significantly reduced. In this study, we analyzed these data using DESeq2 [26] and identified 44 DEGs with an adjusted p-value less than 0.001 (Figure 1D). Five and thirty-nine genes were upregulated and downregulated by hAAT treatment, respectively. Twenty-three of those DEGs were previously identified using Cuffdiff2. Among the 39 down-regulated DEGs, 11 SASP genes were identified (Table 1), including IL-6, IL-8, CCL2, CCL7, IL1Beta, CXCL1, CXCL6, CSF3, CTSK, CXCL5, and CXCL3. Our updated analysis with DESeq2 confirmed that hAAT specifically suppressed the expression of SASP genes in senescent cells.

We then identified the biological processes affected by the hAAT treatment using an EnrichR-based Gene Ontology (GO) enrichment analysis. Fourteen out of the top twenty biological processes ranked, including the top six with the highest scores (Appendix A), were related to inflammation. The nine biological processes with adjusted *p*-value smaller than 1 × 10^−8^ (identified using the GO over-representation test [30]) were highly correlated (Appendix A), indicating these processes shared a common biological activity; i.e., inflammation response. The DEGs correlated with top-ranked biological processes were inflammation-associated genes, most of which were SASPs (Appendix A). These analyses demonstrated that proinflammatory gene expression was significantly and selectively inhibited by hAAT treatment in the senescent cells.

To verify this finding, we next used the KEGG 2019 gene set library for the pathway enrichment analysis. Of the 20 top-ranked pathways affected by hAAT treatment of HCA2 cells, 19 of them were associated with inflammation (Figure 1B). Among these 19 inflammatory pathways suppressed by hAAT, NF-κB played a major regulatory role in 17 of them (Figure 1B). The pivotal role of NF-κB in mediating the observed anti-inflammation effect of hAAT was further confirmed by Transcriptional Regulatory Relationships Unraveled by Sentence-based Text mining (TRRUST), which used literature mining to identify targets of transcription factors and then used that information to pinpoint the transcription factor responsible for regulating a set of genes [35,36]. The TRRUST analysis confirmed that hAAT-downregulated genes were mainly regulated by NF-κB (Rel, RelA, and NKKB1) (Figure 1C,D). Similar results were found in the transcription factor enrichment analysis using Transcription Factor PPIs (Appendix A) and the TRANSFAC and JASPAR PWMs (S1E) gene sets. These analyses strongly indicated that hAAT selectively suppressed NF-κB-targeted SASP genes and inflammatory process.

### 3.2. Human AAT Decreased Nuclear NF-κB Activity before Detectable Changes of Targeted Gene Expressions in Human Senescent Cells

To detect the effect of hAAT treatment on NF-κB activity, nuclear proteins of HCA2 cells treated with fresh medium plus hAAT or PBS were harvested at 3 and 5 days after irradiation (Figure 2A). The results of these experiments showed that hAAT treatment significantly decreased the DNA binding activity (or amount) of RelA (P65) in the nucleus at both day 3 (Figure 2B) and day 5 (Figure 2C). The IL-6 gene expression was significantly lowered by hAAT treatment starting at a later time point, Day 5 (Figure 2D). More interestingly, the hAAT treatment significantly inhibited gene expression of *p65* (Figure 2E) and *p50* (Figure 2F) starting at day 7 and day 5, respectively. In addition, *c-Rel* (Appendix A), as well as the NF-κB associated factors *IκB-α* (Appendix A), *IκB-β* (Appendix A), and *IKK-β* (Appendix A), were also suppressed by hAAT at day 7 in the senescent cells. Together, these results clearly demonstrated that hAAT inhibited NF-κB activity before affecting NF-κB-controlled gene expression. The results showing that hAAT inhibited NF-κB activity, and subsequently *c-Rel* and NF-κB associated factors, were consistent with the notion that NF-κB can positively regulate genes that encode NF-κB subunits (self-regulation) [37,38].

To test the effect of hAAT on postsenescent cells, the irradiated cells were cultured for 7 days without any treatment to allow the senescence process to complete. Then, the postsenescent cells were treated with culture medium plus hAAT or PBS for 5 days (until day 12) and then serum-free medium for 3 days (until Day 15) (Figure 1A). The RT-PCR analysis showed that hAAT significantly decreased *p65* and *p50* gene expression in postsenescent cells starting at day 15 (Figure 3A,B). We noticed that *NFκB2* gene expression in the hAAT-treated group increased on day 12 and decreased on day 15, although there was no statistical difference in comparison with the controls (Figure 3C). Interestingly, the gene expression of *Cathepsin K*, a downstream gene regulated by NF-κB, was inhibited starting on day 12 (Figure 3D). These results again confirmed that the inhibition of hAAT in the gene expressions of NF-κB subunits was not primary (not the first step), but rather a late response (at least later than a downstream gene *Cathepsin K*) in the senescent cells.

### 3.3. Human AAT Inhibited NF-κB Regulator (cIAP-1 and -2) Gene Expression

The noncanonical activation of NF-κB pathways is associated with cellular inhibitor of apoptosis proteins (cIAPs) [39]. Since cIAP-2 was decreased in the hAAT-treated HCA2 cells [25], we next tested the effect of AAT on *cIAP-1* and *-2* gene expression in HCA2 cells during (Figure 1A) and after senescence formation (Figure 2A). The results showed that the hAAT treatment significantly suppressed *cIAP-1* gene expression starting on day 5 during senescence formation (Figure 4A) and on day 15 in postsenescent cells (Figure 4B). *Cellular IAP-2* gene expression was significantly inhibited by hAAT starting on day 12 in the postsenescent HCA2 cells (Figure 4C) (*cIAP-2* gene expression was not detectable during senescence formation; data not shown). More interestingly, transgenic expression of hAAT could suppress *DIAP-1/2* (*cIAP-1/2* analog in *Drosophila*) expression in aged *Drosophila* (Figure 4D,E). These data indicated that hAAT could suppress *cIAP* gene expression in both human senescent cells and *Drosophila*.

Birinapant, a dimeric second mitochondria-derived activator of caspases (SMAC) mimetic, was shown to be able to target cIAP-1 and -2 for degradation [40]. In this study, we used Birinapant to target cIAP-1 and -2 in HCA2 cells by treating the irradiated cells immediately for 7 days. The RT-PCR data indicated that Birinapant treatment significantly decreased *IL-6* gene expression (Appendix A) but did not affect the gene expression of *Cathepsin K* (Appendix A). Birinapant also enhanced *cIAP-2* gene expression (Appendix A) but had no significant effect on *cIAP-1* gene expression (Appendix A), indicating that the inhibition of cIAP activity may increase the demand for cIAP, thus inducing *cIAP* gene expression. Surprisingly, Birinapant also significantly decreased *p65* expression (Appendix A) and showed a trend toward suppressing the gene expression of *p50* (Appendix A). These data, together with the inhibitory effect of hAAT on *cIAPs*, suggested that inhibition of *cIAPs* by hAAT may, at least in part, contribute to decreased SASP and NF-κB gene expression.

### 3.4. Impact of hAAT on the Fitness of Transgenic Drosophila

Similar to the effect in human cells, hAAT also inhibited the expression of *Relish*, the *Drosophila* ortholog of NF-κB [25]. While we previously showed that hAAT driven by Lsp2-Gal4 or fat-body-specific genes with Gal4 reduced aging-associated increases in proinflammatory gene expression and significantly extended the lifespan of transgenic fruit flies [25], it is unknown whether hAAT has any effect on animal fitness. In follow-up studies, we evaluated the impact of hAAT on the fitness of the transgenic flies. First, the locomotor activity was measured as the number of times each fly move crossed a lesser beam, reflecting basic activity level during a 24–48 h period [41]. We found that overall, the hAAT transgenic flies demonstrated moderately increased locomotor activity throughout their lifespans (Appendix A). We next measured the endurance by forcing flies to undergo four negative geotaxis exercises with a 1 min interval and then measured the height of the climbing within 10 s. Interestingly, hAAT transgenic flies had a decreased endurance in this assay. This was true for both female (Appendix A) and male (Appendix A) flies. The difference was mainly restricted to young (days 5 and 11) or middle-aged (day 20) flies. By day 45, the difference was no longer detectable for either sex.

### 3.5. The Anti-Inflammaging Effect of hAAT When Driven by Adult-Specific Gal4

We have shown previously that hAAT driven by Lsp2-Gal4 significantly inhibited inflammaging and extended the lifespan. This effect was also verified in two RU486-activated Gene Switch Gal4 (GS-Gal4) lines [25]. However, in both cases, hAAT was expressed prior to the adult stage. In the case of the GS-Gal4 lines, it was due to leaky expression of the system. In this study, we sought to drive hAAT expression with the Yolk-Gal4 expression system (Figure 5A), which is active specifically in the adult female fat body [42]. We verified the expression of hAAT driven by Yolk-Gal4 and compared it to that driven by Lsp2-Gal4. As expected, we found that during the larval stage, the mRNA level of *hAAT* driven by Yolk-Gal4 was barely detectable (Figure 5B). The protein level of hAAT in the larval homogenate was well below the detection threshold of the ELISA (Figure 5C and Appendix A). In contrast, Yolk-Gal4 drove significantly higher levels of hAAT in the adult stage as compared to Lsp2-Gal4 (Figure 5B,C and Appendix A). Importantly, overexpression of hAAT driven by the adult-stage-only Yolk-Gal4 significantly extended the lifespan of the transgenic flies. A survival analysis was performed for UAS-Background x Yolk-Gal4 (N = 382), UAS-CD4-GFP x Yolk-Gal4 (N = 345), and UAS-hAAT x Yolk-Gal4 (N = 412) to test the antiaging effect of hAAT. Based on a post hoc analysis, the lifespan of the hAAT-expressed flies was significantly longer than that of the background flies (*p* < 2 × 10^−16^) and GFP flies (*p* < 2 × 10^−16^), whereas the two control groups showed a comparable lifespan (*p* = 0.36) (Figure 5D). The lifespan-extension effect of hAAT was confirmed by an independently conducted survival analysis (Appendix A, *p* < 2 × 10^−16^). Similar to what we observed in Lsp2-Gal4-hAAT flies, the lifespan-extension effect of Yolk-Gal4-hAAT *Drosophila* was associated with suppressed gene expression of NF-κB-regulated, proinflammatory antimicrobial peptides, including *dptA* and *dptB* (Figure 5E). These data indicated that transgenic expression of hAAT (mainly in the adult stage) was capable of significantly inhibiting inflammaging and extending the lifespan.

## 4. Discussion

Aging and associated diseases are largely driven by chronic, systemic, low-grade inflammatory processes known as inflammaging [44]. Targeting NF-κB, a transcriptional activator of several proinflammatory pathways, has a significant impact on aging and aging-associated diseases because activation of the NF-κB signaling pathway is associated with the aging process and is responsible for SASP [20,32]. Our previous work indicated that hAAT could extend the lifespan of *Drosophila* and decrease NF-κB-induced SASP [25]. However, the mechanism(s) underlying hAAT functions are not fully understood. In this study, we showed that the first detectable impact of hAAT treatment of senescent HCA2 cells was the inhibition of NF-κB activity, which could in turn play a critical role in the downregulation of SASP genes, NF-κB subunits, and regulators of NF-κB. These results provided a novel avenue for unraveling the mechanisms underlying the widely observed anti-inflammaging effect of hAAT. First, since NF-κB plays an important role in inflammation, these results may be helpful in explaining why hAAT displays a therapeutic effect in many inflammatory disease models [7]. Second, inflammation-related pathways and factors form a complex network that includes positive/negative feedback and self-regulation pathways. It is generally difficult to rule out which one is the key player or target. In the present study, we clearly showed that hAAT treatment decreased NF-κB activity before affecting the expression of other genes.

RNA-seq has been increasingly used in aging studies due to its greater dynamic range and better reliability than older techniques such as microarrays [45,46]. Our previous work identified 42 DEGs (9 upregulated and 33 downregulated) using the CuffLink2 method in senescent HCA2 cells treated with hAAT; of those genes, 8 were SASP genes [25]. In this study, RNA-seq data were analyzed using the DESeq2 method, which identified 44 DEGs (5 upregulated and 39 downregulated). Eleven SASP DEGs were identified using the DESeq2 method, including seven SASP DEGs identified previously. These results confirmed that hAAT significantly suppressed SASP gene expression in the senescent cells. Using RT-PCR as a standard, the DESeq2 method showed a high true positive rate, specificity, positive predictive value, and accuracy [47]. Compared to Cuffdiff2, DESeq2 achieved a higher sensitivity while controlling for the false discovery rate [26]. The DEGs were further used for a pathway and transcription factor enrichment analysis. The results indicated that NF-κB subunits, as well as NF-κB-mediated signaling pathways, were significantly repressed by hAAT treatment in the senescent cells. Several top-ranked pathways or processes have been shown to be inhibited by hAAT in other systems, including the TNF signaling pathway [3,48], autoantibody production [8,9], Toll-like receptor signaling pathway [25], inflammatory response [10], osteoclast differentiation [13] and NF-κB signaling pathway [25]. All these pathways have been shown to contribute to aging and aging-related diseases. The transcription factor enrichment analysis revealed that several NF-κB subunits including Rel, RelA, and NF-κB1 were among the top-ranked transcription factors that were suppressed by hAAT in the senescent cells. Rel and RelA have been shown to be correlated with senescence and aging [49]. One possible limitation of this study was the use of PBS as a control. Protein controls may be used in future studies.

In senescent cells, activation of NF-κB, especially subunit p65, plays a major role in increasing SASP [50]. This clue led us to hypothesize that hAAT may suppress SASP via NF-κB inhibition. Indeed, using an NF-κB binding activity assay, we showed that hAAT could significantly inhibit p65 activity starting at 3 days after X-irradiation during senescence formation, which occurred earlier than the decrease in IL-6 gene expression in hAAT-treated HCA2 cells at day 5. The significant decline in NF-κB subunit gene expressions was also observed later than the activity inhibition with hAAT treatment. These results suggested that the potential primary target of hAAT in senescent cells was NF-κB activity, which consequently led to delayed and decreased downstream IL-6 gene expression, as well as lowered subunits of NF-κB and its regulatory gene expression, possibly via an auto-regulatory loop [37,38]. For postsenescent cells, NF-κB subunit gene expressions were significantly decreased by hAAT treatment 3 days after the withdrawal of hAAT, whereas hAAT suppressed the NF-κB-regulated *Cathepsin K* gene expression immediately after 5 days of treatment. These data also suggested that NF-κB activity was the potentially primary target of hAAT in the senescent cells. However, the mechanism underlying this effect requires further investigation. One possible mechanism is that hAAT, as a circulating protein, interacts with extracellular molecules and indirectly affects NF-κB activity in the target cells. As hAAT can enter the target cells, another possible mechanism is that hAAT may interact with intracellular proteins and directly affect NF-κB activity.

Our previous work suggested hAAT may suppress *cIAP2* gene expression in senescent HCA2 cells [25]. In addition, cIAPs and NF-κB are mutually regulated: (1) cIAPs have been found to be required to activate NF-κB signaling pathways [51,52]; (2) decreasing the gene expressions of *cIAP1* or *cIAP2* in human primary trophoblast cells significantly suppressed the expression of SASP genes, including *IL-6* and *IL-8* [53], the expressions of which were regulated by NF-κB; and (3) NF-κB regulated *cIAPs* transcription [54]. In this study, we tested the change in *cIAPs* induced by hAAT in senescent cells. Since *cIAP2* is undetectable during senescence formation, we only measured the *cIAP1* transcription level. The results showed that cIAP-1 gene expression was significantly inhibited by hAAT starting at day 5 during senescence formation, which was later compared to the suppression of NF-κB activity. These data suggested that the suppression of *cIAP1* gene expression in the hAAT-treated HCA2 cells was likely the consequence of the NF-κB activity repression identified starting at Day 3, which was consistent with the fact that the *cIAPs* could be transcriptionally regulated by NF-κB [54]. These results also led us to exclude the possibility that hAAT inhibited NF-κB activity by regulating the cIAP level during senescence formation. For postsenescent HCA2 cells, hAAT significantly suppressed the transcription level of both *cIAP1* and *cIAP2*. Note that the *cIAP2* gene expression showed a significant decrease immediately after 5 days of hAAT treatment. The role of early onset of a *cIAP-2* decrease requires further study to be demonstrated in the future. More interestingly, consistent with what was observed in the human senescence model, transgenic expression of hAAT in *Drosophila* significantly suppressed *DIAP-1/2* (the *cIAP-1/2* analog in *Drosophila*) at day 35. It was reported that *DIAP-2* was essential to the Relish-mediated IMD inflammatory signaling cascade [55,56], which suggested that suppression of *DIAP-1/2* gene expression may contribute to the hAAT-induced prevention of the aging-related increase in *Relish* and *Diptericin* gene expression [25]. Birinapant is a newly developed SMAC mimetic that can specifically target cIAP-1 and cIAP-2 and leads to their degradation [40]. Birinapant has been widely used as a potential therapeutic molecule to treat cancer [57,58]. In this study, Birinapant (10 µM; LC Laboratories) was added to irradiated HCA2 cells and showed an inhibitory effect on *IL-6* and *p65* gene expressions after 7 days of treatment. While significantly decreasing the cIAP protein level (at 10 µM) [40], Birinapant treatment in senescent cells induced an elevation in the mRNA of *cIAPs*, which could have resulted from negative loop regulation. Nonetheless, these data suggested that Birinapant may act as an inhibitor of inflammaging.

Using the Yolk-Gal4 line, in which the transgene expression is restricted to the adult stage, we demonstrated that the anti-inflammaging effect of hAAT on transgenic *Drosophila* was independent of the possible effect on animal development. In this work, we also showed that the anti-inflammaging effect of hAAT in *Drosophila* was associated with an enhancement of locomotor activity, suggesting hAAT not only mitigated the NF-κB-mediated aging process, but also slowed the aging-associated decline in locomotor activity. The decreased endurance observed for hAAT-expressing flies is intriguing. It is suggestive of possible interlink and balance between metabolism and innate immunity.

Our study strongly indicat that hAAT controls an innate immune mechanism that is highly conserved. It is highly plausible that the *Drosophila* genome contains one or more serine protease inhibitors (serpins) that is (are) the functional ortholog(s) of hAAT. Unfortunately, such ortholog(s) could not be identified by a sequence comparison alone because many of the 28 serpin genes in the fruit fly genome possess a strong sequence similarity with hAAT in the highly conserved serpin domain (InterPro ID-IPR023796), which accounts for 80–90% of the total length of these serpins. A functional screen is currently being conducted to reveal the ortholog(s) of hAAT in *Drosophila*.

## 5. Conclusions

In conclusion, our results provide a novel insight into the functional mechanism of hAAT with regard to its anti-inflammaging effect (graphical abstract). We showed that AAT inhibits NF-κB activity prior to downregulation of proinflammatory gene expressions, suggesting that the anti-inflammaging effect of hAAT is mainly though the inhibition of NF-κB. In addition, we show that AAT intervention starting at the adult stage is sufficient to suppress inflammaging, implicating a potential role of AAT treatment in aging-related diseases.

## Figures and Tables

**Figure 1 biomolecules-12-01347-f001:**
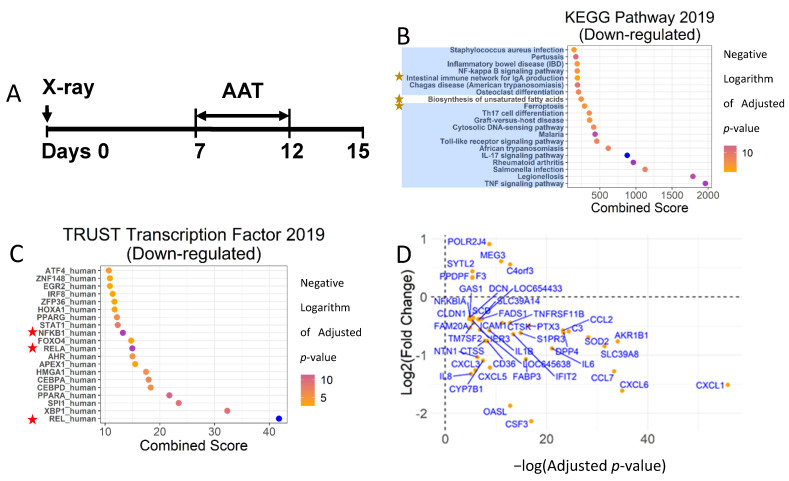
Human AAT suppressed NF-κB-mediated inflammatory pathways in senescent HCA2 cells. (**A**) Study strategy: the HCA2 cells were irradiated and cultured without any interference for 7 days and then treated with hAAT (2 mg/mL) or 1× PBS containing fresh medium for 5 days. At day 12, FBS-free medium was used to replace the hAAT- or PBS-containing medium and culture the cells for 3 days. Total RNA was extracted at day 12 or 15 and processed for RNA-seq or PCR. (**B**) Pathway enrichment analysis using KEGG Pathway 2019 Gene Set. Blue shade: pathways related to inflammation; yellow star: pathways not regulated by NF-κB. (**C**) Transcription factor enrichment analysis using TRUST Transcription Factor 2019 Gene Set. Red star: NF-κB subunits. (**D**) The DEGs were identified by an adjusted *p*-value less than 0.01. The log2 (fold change) of DEGs were plotted against-log (adjusted *p*-value). The combined score was the combination of the Fisher’s exact test *p*-value and deviation from the expected rank of each term. The *p*-value, adjusted *p*-value, and combined score were calculated following [30].

**Figure 2 biomolecules-12-01347-f002:**
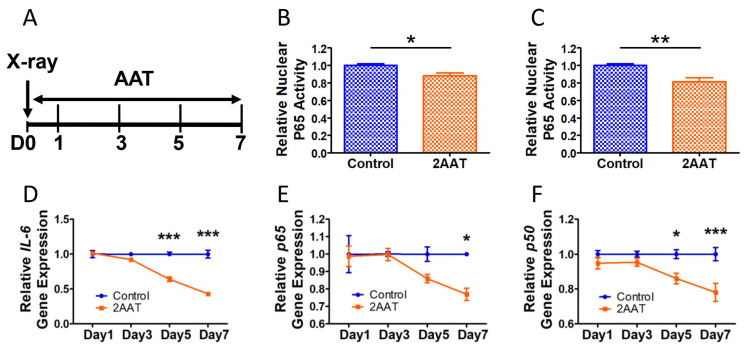
Human AAT inhibited NF-κB activity before affecting its gene expression in senescent HCA2 cells. (**A**) Study strategy: the HCA2 cells were irradiated and immediately treated with either hAAT (2 mg/mL) or 1× PBS containing fresh medium. The mRNA or nuclear extraction were conducted at 1, 3, 5, and 7 days after irradiation. (**B**) Relative nuclear p65 activity 3 days after irradiation (N = 4, representative data from 2 experiments). (**C**) Relative nuclear p65 activity 5 days after irradiation (N = 6, representative data from 2 experiments). Gene expression of *IL-6* (**D**), *p65* (**E**), and *p50* (**F**). Error bar represents standard error of the mean. N = 4 for all gene expression data. A *T*-test was used for statistical analysis. * *p* < 0.05, ** *p* < 0.01, *** *p* < 0.001.

**Figure 3 biomolecules-12-01347-f003:**
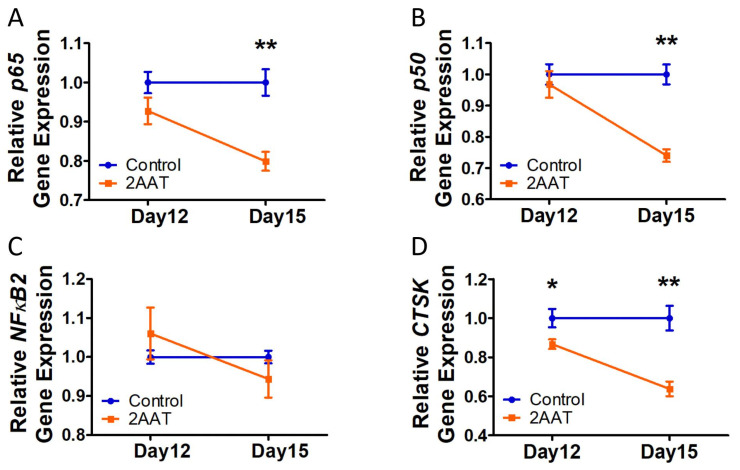
Human AAT treatment decreased the NF-κB gene expression after affecting its downstream gene expression in postsenescent HCA2 cells: (**A**) *p65* gene expression; (**B**). *p50* gene expression; (**C**). *NF-κB2* gene expression; (**D**). *Cathepsin K* gene (NF-κB downstream gene) expression. Error bar represents standard error of the mean. N = 5 for day 12 and N = 3 for day 15. A *T*-test was used for statistical analysis. * *p* < 0.05, ** *p* < 0.01.

**Figure 4 biomolecules-12-01347-f004:**
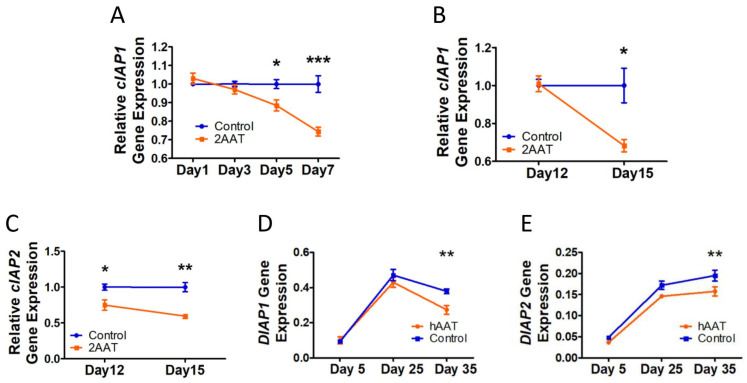
Human AAT suppressed cIAP-1/2 gene expression in senescent HCA2 cells and aged *Drosophila*: (**A**) *cIAP-1* gene expression during senescence process (N = 4); (**B**) *cIAP-1* gene expression in postsenescent HCA2 cells (N = 5); (**C**) *cIAP-2* gene expression in postsenescent HCA2 cells (N = 3). In (**A**–**C**), 2AAT indicates that 2 mg/mL hAAT was used. (**D**) *DIAP-1* gene expression following aging in transgenic flies; (**E**) *DIAP-2* gene expression following aging in transgenic flies. In (**D**,**E**), hAAT indicates data from hAAT transgenic lines and control indicates data from control lines. Five to twelve flies were processed as one replicate. Three replicates were used for *DIAP-1/2* analysis. Error bar represents standard error of the mean. A *T*-test was used for statistical analysis. * *p* < 0.05, ** *p* < 0.01, *** *p* < 0.001.

**Figure 5 biomolecules-12-01347-f005:**
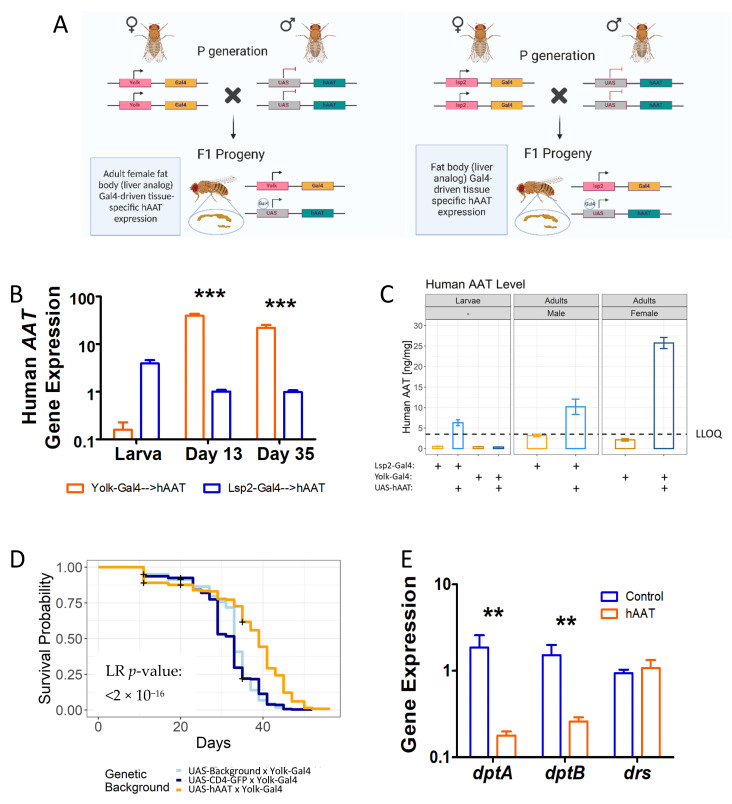
The anti-inflammaging effect of hAAT when only expressed in adult *Drosophila*. (**A**) Schematic of the *Drosophila melanogaster* adult fat body Yolk-Gal4-driven hAAT expression system. (**B**) Whole-body Yolk-Gal4-driven hAAT mRNA transcription across the fly lifespan obtained using qRT-PCR (N = 5–6). Relative expression was calculated using the pflaffl method [43] with *Rpl32* as a reference gene; a two-way ANOVA test was used for statistical analysis (*** *p* < 0.001). (**C**) Yolk-Gal4-driven hAAT secretion detected using a hAAT-specific ELISA across the fly lifespan. A mixture of stage 2 and stage 3 larva was used. Bar represents the mean hAAT amount per fly. Error bar represents the standard error. Dotted line represents lower limit of quantitation (LLOQ). (**D**) Effect of Yolk-Gal4-driven hAAT expression on lifespan in female *Drosophila melanogaster* (N = 382, 345, and 412 at the beginning of the experiment for Background (docking strain) x Yolk-Gal4, UAS-CD4-GFP x Yolk-Gal4, and UAS-hAAT x Yolk-Gal4, respectively). A log-rank test was used for statistical inferences. The plus sign represents data censoring. (**E**) Effect of Yolk-Gal4-driven hAAT expression of *Diptericin A* (*dptA*)and *B*
**(***dptB***)** in female adult *Drosophila melanogaster* aged to 35 days. Relative expression was calculated using the Pflaffl method [43] with *Rpl32* as a reference gene; a Mann-Whitney test was used for statistical analysis. ** *p* < 0.01. Error bar represents standard error of the mean.

**Table 1 biomolecules-12-01347-t001:** SASP genes identified as differentially expressed genes (DEGs) by DSEq2. The SASP genes with a blue shade were also identified using the Cuffdiff2 method in our original paper [25]. The *p*-value was calculated based on fold change using a Wald test. The BH method was used to control the false discovery rate.

DEG	Log2 (Fold Change) for DESeq2	Cuffdiff2		DESeq2	
		*p*-Value	*Q*-Value	*p*-Value	Adjusted *p*-Value
CXCL1	−1.50957	5.00 × 10^−5^	0.00154	4.92 × 10^−29^	5.87× 10^−25^
CXCL6	−1.6105	5.00 × 10^−5^	0.00154	1.12 × 10^−19^	6.71 × 10^−16^
CCL7	−1.27657	5.00 × 10^−5^	0.00154	1.10 × 10^−18^	3.27 × 10^−15^
CCL2	−0.5778	5.00 × 10^−5^	0.00154	6.18 × 10^−14^	8.19 × 10^−11^
IL6	−0.88616	5.00 × 10^−5^	0.00154	6.04 × 10^−13^	7.21 × 10^−10^
CSF3	−2.13496			4.00 × 10^−11^	4.34 × 10^−8^
CTSK	−0.44707			2.12 × 10^−8^	1.33 × 10^−5^
CXCL5	−1.2117			2.61 × 10^−7^	0.000148
IL1B	−0.57528	5.00 × 10^−5^	0.00154	3.25 × 10^−7^	0.000169
CXCL3	−1.25985			7.44 × 10^−6^	0.00269
IL8	−1.32121	5.00 × 10^−5^	0.00154	2.09 × 10^−5^	0.006219

## Data Availability

The RNA-seq datasets generated and/or analyzed during the current study are available in the GEO repository (https://www.ncbi.nlm.nih.gov/geo/query/acc.cgi?acc=GSE183948, accessed on 18 August 2022). The other datasets used and/or analyzed during the current study are available from the corresponding author upon reasonable request.

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
