# Peer review of "Human Alpha 1 Antitrypsin Suppresses NF-κB Activity and Extends Lifespan in Adult Drosophila"

_biomolecules, 2022, doi:10.3390/biom12101347_

Round 1

Reviewer 1 Report

This research investigates the anti-inflammation mechanisms of human alpha 1 antitrypsin in senescent HCA2 cell and transgenic Drosophila models, and it clearly revealed that the NF-kB activity and NF-kB controlled genes were down-regulated or inhibited by hAAT treatment, thus leading to the anti-inflammatory effect. The experiments were thoroughly designed using multiple approaches, and data were carefully interpretated and presented.

There are a few minor comments/suggestions to be addressed prior to publication:

1. In Fig 1A, the cells were treated with hAAT or PBS for 5 days after day 7. My understanding is that the cells were cultured with the culture medium plus hAAT for 5 days, so the proper control for without hAAT should be culture medium instead of PBS? This question applies to other experiment treated with hAAT as well.

2. In Fig 2, for those who are not familiar with NF-kB and NK-kB inflammatory pathway, it may be helpful to include another Figure in the supplementary to illustrate NF-kB, subunits and pathway.

3. In Fig 3, did the author test the IL-6 expression level as well?

4. Figure 5A is not clear and needs higher resolution.

Author Response

Dear Reviewer,

First, we want to thank you for your comments and suggestions. As suggested, we have revised our manuscript. The attached file is our response to your point. 

Reviewer 2 Report

General

·        In this paper by Yuan, et al. the authors have conducted an interesting study aimed at assessing a the association between hAAT, NFkP, inflammation and senescence.

·        The manuscript is well-written and the methodology is well-thought. The data is presented in a flowing manner which assists in the understanding of the research process and the thought behind it. It was pleasant and interesting to read. Thank you.

·        Please proof the paper in order to maintain a coherency in abbreviation of hAAT and AAT (e.g. hAAT in line 34 and AAT in line 37, 42 etc…).

·        The manuscript is well-referenced with relevant and updated manuscripts.

Introduction

·        The introduction section of the manuscript is well written and gives sufficient information to understand the main topics tackled in the manuscript.

·        I would suggest giving readers slightly more information about the Drosophila model in order to allow better understating and appreciation of the impressive results presented later. However, this is only a suggestion and not mandatory.

·        The introduction of this paper could significantly benefit from further discussion on other qualities observed in hAAT. Among which are:

o   ROS scavenging (especially in this context as ROS is are a marker or possibly inducer of senescence).

o   hAAT ability to transfer from extracellular to intracellular compartments, as previously shown in relation to NFkB (32795666). This is of special relevance to this specific manuscript as the effects of extracellular hAAT are suggested to be mediated by the intracellular NFkB.

o   Further elaboration on the immunomodulative and anti-apoptotic effects of hAAT such as in the case of T1DM and pancreatic islets.

Materials and Method

·        General

o   This section is written in a concise and efficient manner.

o   What was the source of hAAT in this research? Recombinant? Serum-purified?

o   While PBS may represent a sufficient control, in future study consider using a more appropriate one such as purified albumin as it represents a protein addition and is of similar molecular size to hAAT.

·        The methodology of cell culture and x-irradiation should be more elaborate:

o   Following irradiation, was the media replaced to avoid cellular and ROS interference?

o   Were PBS or hAAT added to existing MEM media?

o   Was hAAT replenished at any point during the 7 days to account for degradation?

o   Was culture media replaced at any time during the experiment?

·        RNA extraction and qPCR

o   I’m unfamiliar with housekeeping genes in HCA2 cells undergoing inflammatory process. While GAPDH is well known for its stability in keratinocytes there were reports of transcriptional changes in inflammation. Have the authors assessed at least one more housekeeping gene (e.g. PGK1)?

·        Senescence

o   While the irradiation model is indeed familiar for induction of cellular senescence, was its presence validated in the authors’ systems (e.g. Beta gal, FDG, Ki-67 loss, or even ROS expression?)

o    

·        Statistical analysis

o   Given small sample sizes, please use the Mann-Whitney test and not t-test.

Results

·        General

o   This section of the manuscript is written in a descriptive yet flowing manner which is pleasant to read. The paragraphical division of the section is well-thought and allows for understanding of the research process.

o   In this section as a whole the authors often use the term 'gene expression' but describe gene transcription. Please change the phrasing to be more appropriate.

·        Section 2.2

o   Section 2.2 could benefit from proofing and language editing.
For example (there is more to be improved), please change the phrasing from “the nuclear protein in HCA2 cells treated with or without hAAT was harvested 3 and 5-day post irradiation” to “nuclear proteins of HCA2 cells, treated with or without hAAT, were harvested at 3- and 5-days post irradiation.”

o   Please change the P65 in figure 2 to amount and not activity as it wasn’t assessed.

o   In Figure 3C one can see that NFkB2 transcription is increased at day 12 (day +5 after allegedly induced senescence), albeit in a non-significant manner. However, the authors state that hAAT didn't affect the transcription of NFkB2. Please state this effect in the text and discussion as hAAT has been shown in several publications to expedite the timeline of inflammation.

·        Sections 2.4-2.5

o   This is very impressive. However, I have to admit I'm unfamiliar with these types of experiments, and thus unable to suggest improvements or alterations to this segment.

o   Regarding Figure 5D, please state in the figure as well the groups between which the LR test was performed and found significant.

Discussion

General notes:

·        Please include in your limitation the fact the hAAT treatment was compared to PBS and not a protein-control such as albumin.

·        While beautifully displaying a wide array of effects induced by hAAT over NFkB, the authors should discuss the discrepancy between hAAT being a large extracellular protein and the fact that NFkB is an intranuclear protein.

Author Response

(The authors gave the same response as above.)
